

# Can non-destructive DNA extraction of bulk invertebrate samples be used for metabarcoding?

Melissa E. Carew[1], Rhys A. Coleman[2] and Ary A. Hoffmann[1]

[1] School of BioSciences, The University of Melbourne, Melbourne, Victoria, Australia
[2] Applied Research, Melbourne Water Corporation, Melbourne, Victoria, Australia

## ABSTRACT

**Background**. High throughput DNA sequencing of bulk invertebrate samples or metabarcoding is becoming increasingly used to provide profiles of biological communities for environmental monitoring. As metabarcoding becomes more widely applied, new reference DNA barcodes linked to individual specimens identified by taxonomists are needed. This can be achieved through using DNA extraction methods that are not only suitable for metabarcoding but also for building reference DNA barcode libraries.
**Methods**. In this study, we test the suitability of a rapid non-destructive DNA extraction method for metabarcoding of freshwater invertebrate samples.
**Results**. This method resulted in detection of taxa from many taxonomic groups, comparable to results obtained with two other tissue-based extraction methods. Most taxa could also be successfully used for subsequent individual-based DNA barcoding and taxonomic identification. The method was successfully applied to field-collected invertebrate samples stored for taxonomic studies in 70% ethanol at room temperature, a commonly used storage method for freshwater samples.
**Discussion**. With further refinement and testing, non-destructive extraction has the potential to rapidly characterise species biodiversity in invertebrate samples, while preserving specimens for taxonomic investigation.

Corresponding author
Melissa E. Carew,
mecarew@unimelb.edu.au

## INTRODUCTION

Species identification using high throughput DNA sequencing (HTS) of bulk samples containing multiple species (metabarcoding) is being increasingly applied in environmental monitoring, as it enables rapid identification of a wide range of taxa (*Yu et al., 2012*; *Zhou et al., 2013*; *Gibson et al., 2014*; *Gibson et al., 2015*; *Elbrecht et al., 2017*). However, associating sequences generated by metabarcoding with species names is dependent on comprehensive DNA barcoding libraries where individual specimens identified by taxonomists are DNA barcoded (*Hajibabaei et al., 2011*; *Baird & Hajibabaei, 2012*; *Carew et al., 2013*; *Zimmermann et al., 2014*). Some taxonomic groups and geographic regions have comprehensive DNA barcode libraries (e.g., *Hebert et al., 2013*; *Hendrich et al., 2015*; *Hebert et al., 2016*), but many taxonomic groups or taxa from other geographical regions lack such

coverage (see *Carew et al., 2017*; *Kranzfelder, Ekrem & Stur, 2017*). This means that when environmental samples containing invertebrates are processed using metabarcoding, many sequences cannot be identified to species. This does not prevent the use of DNA barcodes for environmental monitoring, as sequences can be grouped into molecular operational taxonomic units (MOTUs) (*Blaxter et al., 2005*), which can be associated with various environmental parameters (i.e., *Pawlowski et al., 2014*; *Lanzén et al., 2016*). However, it does mean that pre-existing taxonomic and environmental information based on other approaches cannot be easily integrated (*Carew et al., 2007*; *Schafer et al., 2011*). Furthermore, an MOTU approach prevents the accuracy of DNA barcodes for species identifications to be tested (*Lee, 2004*; *Moritz & Cicero, 2004*), and it complicates the detection of PCR artifacts and chimeras when analyzing metabarcoding data (*Creer et al., 2010*; *Carlsen et al., 2012*).

To ensure that reference DNA barcode libraries are developed for invertebrate samples that are processed with HTS, field samples can be taken in duplicate, where one is used for creating local reference DNA barcodes based on individual specimens and the other is homogenized and used for metabarcoding (e.g., *Gibson et al., 2015*). This approach can lead to a high species detection rate with HTS (*Gibson et al., 2014*; *Gibson et al., 2015*). However, it is possible that some taxa, particularly rare species, will not be found in both samples. If these species are only detected in the metabarcoded sample, there is not an opportunity to produce a reference DNA barcode when specimens are destroyed during DNA extraction.

Where only single samples are available, dissecting tissue or legs from individuals used for HTS keeps specimens largely intact for individual reference DNA barcoding. Detailed photographs of individual specimens can further assist identification, especially where damage occurs during dissection (e.g., *Sweeney et al., 2011*). However, this is a slow and difficult process, particularly for small taxa such as water mites, or for taxa with tissues inside shells, such as bivalves and gastropods. The preservation ethanol can also be used for non-destructive metabarcoding, leaving specimens undamaged (e.g., *Hajibabaei et al., 2012*). However, this approach can be less successful than tissue-based extraction for metabarcoding, depending on ethanol quality or re-use, storage time, presence of PCR inhibitors, as well as species biomass and composition (*Hajibabaei et al., 2012*; *Carew et al., 2013*).

A non-destructive DNA extraction method that enables reliable detection of taxa in bulk samples would help to overcome some of these challenges. Non-destructive methods involving temporarily immersing whole specimens in an extraction buffer have been successfully applied to individual specimens for reference DNA barcoding (*Rowley et al., 2007*; *Castalanelli et al., 2010*; *Porco et al., 2010*; *Krosch & Cranston, 2012*; *Wong et al., 2014*; *Cornils, 2015*). However, they have not been tested with bulk invertebrate samples, like those used for metabarcoding. It is not clear whether DNA from the same individuals extracted non-destructively for metabarcoding can then be used for creating reference DNA barcodes. Also, it is not known whether combining species from different taxonomic groups with varying body forms in a non-destructive DNA extraction can be successfully metabarcoded to determine species biodiversity.

In this study, we aim to test whether it is possible to apply non-destructive DNA extractions to bulk invertebrate sample intended for metabarcoding. We first examine if individual invertebrates from common freshwater orders can be successfully extracted for metabarcoding and then reference DNA barcoding. We subjected taxa to a non-destructive total genomic extraction using a commercial kit, and followed with a Chelex extraction to test whether the same specimens can provide adequate DNA after multiple extractions. We then inspected the morphological integrity of specimens for taxonomic work after non-destructive extraction. Next, we constructed multiple bulk invertebrate samples for metabarcoding to compare species detection rates of samples processed using non-destructive DNA extractions to two different tissue-based DNA extraction methods. Finally, we tested the non-destructive DNA extraction method on samples collected for routine bioassessment, but stored in different ethanol concentration and temperatures, to determine if non-destructive extraction could be applied to routinely stored invertebrate collections.

## MATERIAL & METHODS

### Component 1: non-destructive DNA extractions on individuals

The first component of this study aimed to determine whether a non-destructive DNA extraction can be used to obtain DNA from whole individuals from common freshwater macroinvertebrate orders/subclasses (Fig. 1A). To test this, we included a small experiment (Experiment 1a) that examined the incubation time required for specimens to be immersed in extraction buffer to obtain DNA without destruction of morphological characters, followed by a second broader experiment (Experiment 1b) to see how the method performed on individuals from multiple families covering 15 common orders/classes/subclasses.

A total genomic extraction (Extraction 1; Fig. 1A), which is commonly used to prepare samples for HTS (e.g., *Elbrecht, Peinert & Leese, 2017*), was used to obtain DNA from whole and intact invertebrate specimens. For extraction, ethanol was removed using a pipette from microcentrifuge tubes containing individual specimens, then 180 µL T1 buffer and 25 µL of proteinase K (extraction buffer) from the Nucleospin DNA extraction kit (Macherey-Nagel Inc. Bethlehem, PA, USA) were added. We tested the whether the presence of ethanol interfered with the DNA extraction by soaking several specimens overnight in TE buffer prior to extraction, but found no difference in successful amplification between soaked specimens and those with ethanol removed just prior to extraction (Table S1). Specimens were then incubated in the extraction buffer at 56 °C. Based on other studies (*Rowley et al., 2007*; *Castalanelli et al., 2010*; *Porco et al., 2010*), we incubated specimens for 1 hr in the extraction buffer, but also tested a subset of taxa for 30 mins and/or 3 hrs depending on their levels of sclerotization (Tables S1 and S2). The extraction buffer was removed and placed into a new 1.5 ml microcentrifuge tube for DNA extraction using a Nucleospin DNA extraction kit (Macherey-Nagel Inc. Bethlehem, PA, USA) following the manufacturer's instructions. After extraction buffer removal, absolute ethanol was added

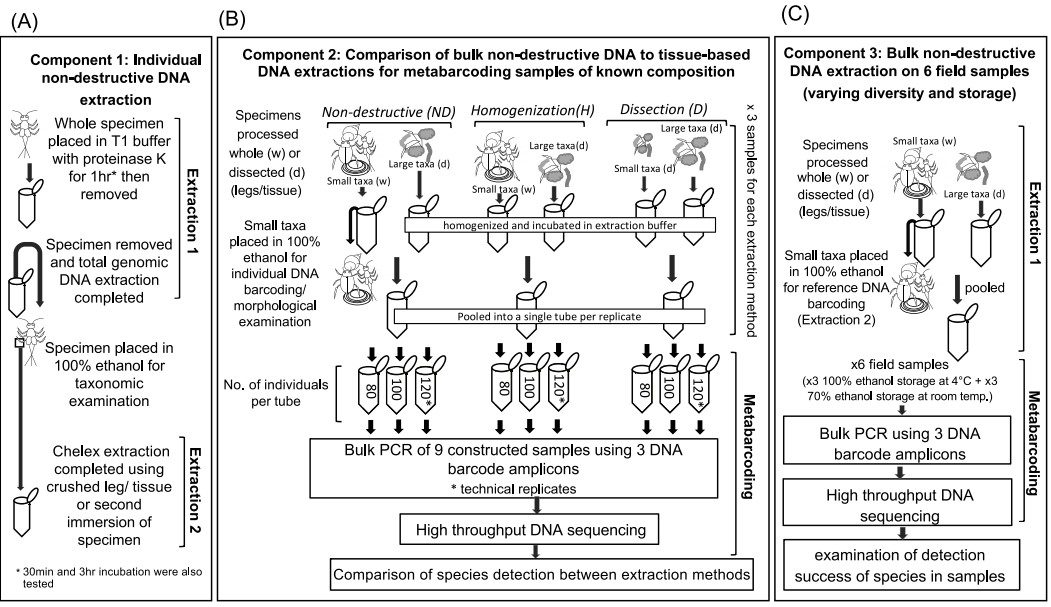

**Figure 1** **Diagram of the workflow for testing the success of non-destructive DNA extraction.** (A) Individual non-destructive DNA extraction (Extraction 1) followed by a Chelex extraction (Extraction 2) performed on single specimens from 15 different macroinvertebrate orders (Component 1). (B) Comparison of the detection of taxa via metabarcoding by using three DNA extraction based on non-destructive processing, homogenization of whole small taxa and dissection of all taxa (Component 2). (C) Comparison of the detection of taxa via metabarcoding from six field collected samples stored in different ethanol concentration, and temperatures an with varying levels of diversity. Drawings by Melissa Carew.

to microcentrifuge tubes containing whole specimens to preserve them for subsequent Chelex extraction and taxonomic identification. The condition of specimens was inspected under a dissecting microscope (Leica Microsystem and Instruments, Wetzlar, Germany).

A Chelex extraction (Extraction 2; Fig. 1A) was then performed to establish if two extractions of the same specimen was possible, i.e., an initial extraction intended for metabarcoding followed by an extraction intended for individual DNA barcoding. The material used for Chelex extraction varied among taxonomic groups, depending on how specimens needed to be prepared for identification (Table S2) but followed the methods of *Carew, Pettigrove & Hoffmann (2003)*.

The success of both DNA extractions was determined by amplification of the DNA barcode region (*Hebert et al., 2003*) using the primer set HCOI2198/LCOI1490 from (*Folmer et al., 1994*) according to *Carew & Hoffmann (2015)*, as this primer set was known to amplify the majority of taxa considered in the study. Some DNA barcode amplicons isolated with DNA extraction methods were sequenced to ensure that the intended species were extracted. Sequencing reactions were performed in both directions using an ABI 3730XL capillary sequencer (Applied Biosystems, Foster City, California, USA) with sequencing reactions and runs performed by Macrogen (Seoul, Korea).

After DNA extraction, taxa were identified to the lowest possible level using relevant taxonomic keys (see *Hawking, 2000*). In most instances this was species, but for some

groups we had difficultly identifying species due to a lack of keys, expertise or because specimens were immature or damaged (e.g., most Ephemeroptera were missing legs/cerci prior to being processed).

All macroinvertebrate specimens used for these experiments were collected using a 250 μm net from riffle and pool habitats from several streams/rivers around the greater Melbourne area (Victoria, Australia) and stored in 100% ethanol at −20 °C until required.

## Component 2: bulk DNA extraction for metabarcoding

In the second component of this study, we examined bulk DNA extraction using the non-destructive (ND) extraction protocol above (as used on individuals) and compared it to two commonly used tissue-based extraction methods based on total homogenization (H) and dissection of tissues (D) (Fig. 1B). To allow a comparison between the three extraction methods (ND, H and D), we constructed three sets of invertebrate samples each containing three 'replicates'. We choose to use high numbers of individuals and a species composition representative of what we have previously found in field samples (e.g., *Carew et al., 2018*) to construct invertebrate samples. The first set contained 80 individuals (Sample A), the second set 100 individuals (Sample B) and third set contained 120 individuals (Sample C) (Fig. 1B). One 'replicate' from each set was used to compare metabarcoding success of the ND, H and D based extraction methods. Technical replicates (i.e., metabarcoding the same DNA extraction twice) were performed on the third set containing 120 individuals to examine the robustness of species detection. Unfortunately, it was not possible to make identical replicates given that we could not mount or destroy samples to check species identification. Instead, the composition of each replicate was standardized as much as possible by selecting taxonomic groups that could be identified to a low taxonomic level under a dissecting microscope, and/or specimens that had been collected from the same sites. We also used similar sized individuals for each taxon to control for differences in biomass. Not all taxonomic groups considered individually in component 1 could be used in constructing samples due to limited availability of material, but we tried to represent animals of varying levels of sclerotization and size.

When constructing samples, we focused on testing how a non-destructive method could be applied to small taxa only ($<7$ mm$^2$) in bulk DNA extractions; all large taxa ($>7$ mm$^2$) were dissected when comparing extraction methods to reduce the likelihood that larger species affected the detection of overall species diversity (*Elbrecht, Peinert & Leese, 2017*). Furthermore, larger taxa tend to be less common and/or more easily dissected than smaller taxa in bulk samples, making this a practical approach.

When comparing the three extraction methods, smaller taxa were processed differently. For completely homogenized samples (H), dissected tissue of large taxa and complete specimens of smaller taxa were combined for DNA extraction. For dissected samples (D), all taxa (both large and small) were dissected and then legs/tissues were combined and homogenized for DNA extraction. Dissected taxa were returned to absolute ethanol. For the non-destructive (ND) approach, absolute ethanol was removed from tubes containing small taxa using a pipette and 180 μL of buffer T1 with 25 μL proteinase K from the Nucleospin DNA extraction kit (Macherey-Nagel Inc.) was added, ensuring specimens

were completely immersed. Tubes were incubated for 1 hr at 56 °C. After incubation, specimens were placed back into absolute ethanol. The tissues/legs dissected from the larger taxa were homogenized separately. All homogenization steps were performed in 1.5 ml microcentrifuge tubes. Tissues were snap frozen in liquid nitrogen and crushed with a sterile pestle. Then 180 μL of the T1 buffer and 25 μL proteinase K were added, and samples incubated for 3 hrs at 56 °C. After incubation all samples were DNA extracted using a Nucleospin tissue kit (Macherey-Nagel Inc.) following the manufacturer's instructions. DNA was eluted in 100 μL of the elution buffer from the Nucleospin tissue kit. The time taken to prepare material for each DNA extraction method and the number tubes used in extractions was recorded. Where multiple extraction tubes were used for a sample, DNA extractions were combined prior to HTS.

A subset of the individuals, representing different taxonomic groups used in bulk non-destructive extraction, was then subjected to individual DNA extraction (see Table S3), DNA barcode amplification and sequencing according to the methods described for extraction 2 (above).

## Component 3: non-destructive DNA extraction of field-collected samples

In the final component, we examined whether the non-destructive DNA extraction method can be used on samples collected for morphological identification as part of rapid bioassessment surveys (Fig. 1C). We obtained six macroinvertebrate samples collected on two occasions (spring 2014 and autumn 2016) from three sites along the Merri Creek, Melbourne, Australia (Table S4). Prior to metabarcoding, the six samples were identified mostly to family level according to the Rapid Bioassessment Protocols developed by EPA Victoria (available at: http://www.epa.vic.gov.au/~/media/Publications/604%201.pdf). Therefore, family level information was available for comparison with our species level DNA barcode identifications. Sites were selected based on the diversity of macroinvertebrate families: Coburg Lake (MCL) had low macroinvertebrate diversity, Rushwood Dr (MRD) had high diversity, while the third site at O'Herns Rd (MOH) had an intermediate level. Prior to metabarcoding, spring 2014 samples were stored at room temperature in 70% ethanol, while the autumn 2,016 samples were stored in 100% ethanol at 4 °C.

For metabarcoding, we followed the non-destructive DNA extraction workflow for bulk DNA extraction outlined in component 2. Specimens in samples representing taxa not found with metabarcoding or had not been previously DNA barcoded were removed from samples and individually DNA barcoded.

## High throughput DNA sequencing of bulk invertebrate samples

A two-step PCR process was used to obtain amplicons for Illumina MiSeq sequencing. The first PCR involved amplifying the DNA barcode region (Hebert et al., 2003) using three PCR primer sets. The primer sets include LCOI1490 (Folmer et al., 1994)/MLepR2 (Hebert et al., 2013); B (Hajibabaei et al., 2012)/COIBrev (5′-GATCARACAAAYARWGG YATWCGRTC-3′) and miCOIintF (Leray et al., 2013)/HCOI2198 (Folmer et al., 1994). Primers were selected based on their ability to amplify a broad range of taxa. While some

taxa may be missed in metabarcoding depending on the primers selected, using the same primer pairs across all extraction methods allowed a comparison of the extraction methods relative success for detecting species.

First round PCR reactions contained 2 µL of DNA template, 16.4 µL molecular biology grade water, 2.5 µL PCR buffer (Invitrogen, Carlsbad, CA, USA), 1 µL MgCl$_2$ (50 mM), 2 µL dNTPs mix (25 mM of each dNTP), 0.5 µL forward primer (10 µM), 0.5 µl reverse primer (10 µM), and 0.1 µL Platinum Taq polymerase (5 U/ml) (Invitrogen, Carlsbad, CA, USA) in a total volume of 25 µL, and were amplified in triplicate using the PCR conditions from *Hajibabaei et al. (2012)*. The PCR replicates were pooled and cleaned using a Mag-Sera magnetic beads (GE Healthcare Australia, Sydney, Australia). Cleaned amplicons were quantified with a Qubit Fluorometer using the dsDNA HS Assay Kit (Life Technologies, Carlsbad, CA, USA).

Template-specific primers had Illumina adaptors incorporated onto the 5′ end for the attachment of Nextera-XT Illumina indexes (Illumina Corporation, San Diego, CA, USA) in the second round of PCR. The second round PCRs were performed after pooling three amplicons in approximately equal molar ratios. Reactions used 15 µL of the first-round amplicons, 25 µL BIO-X-ACT short mix (Bioline, London, England), 5 µL forward Nextera-XT index primer (10 µM), and 5 µL reverse Nextera-XT primer (10 µM). PCR conditions were as follows: 94 °C for 5 mins followed by 12 cycles of 94 °C for 30 s, 55 °C for 30 s, 72 °C for 30 s, then 1 cycle of 72 °C for 5 mins. All amplicons were then cleaned again with Mag-Sera magnetic beads (as above).

Library quantification, normalization, pooling and the Illumina MiSeq run using a 600-cycle flow cell MiSeq sequencing kit V3 (300 bp × 2) (Illumina, San Diego, CA) were performed by Australian Genome Research Facility Ltd (AGRF) according to the manufacturer's protocols. Raw sequence data generated with HTS was deposited in the National Center for Biotechnology Information (NCBI) Short Read Archive (SRA) database under BioProject PRJNA413851.

## HTS bioinformatics analysis

Sequences from HTS were imported into Geneious version R10 (*Kearse et al., 2012*) and were trimmed to <197 bp. FLASH version 1.2.9 (*Magoc & Salzberg, 2011*) was used to merge set paired reads with default settings. Merged reads were annotated with each set of forward and reverse primer pairs and then extracted to isolate each of the three amplicons. Only amplicons of appropriate size (± 6 bp of the amplicons expected size) with both forward and reverse primers on each end were retained. Primers were then trimmed and a custom *de novo* assembly at 98% similarity was used to reduce redundancy in the dataset. The number of reads that contributed to each contig (group of sequences with >98% sequence similarity) from the *de novo* assembly was recorded. Singletons and contigs containing ten or fewer reads were discarded (see *Bokulich et al., 2013*). The remaining contigs were aligned, edited and checked for an open reading frame. Contigs were then BLAST searched against a DNA barcoding reference database of freshwater macroinvertebrates, and species with >97.5% match were identified. The reference database of freshwater macroinvertebrates was constructed by downloading and combining DNA

**Table 1** Outline for the experimental workflow for testing non-destructive DNA extraction.

| Component | Questions | Answers |
|---|---|---|
| 1. Non-destructive DNA extraction of single individuals | Can non-destructive DNA extraction be used to obtain DNA from macroinvertebrates? | Yes, but not from taxa with little sclerotization, such as the Oligochaeta and Hirudinea. |
| | What is a suitable incubation time for immersion in extraction buffer that does not destroy morphological traits but yields DNA? | An hour works best for most taxa, but for Amphipoda less time would be needed to avoid damage to taxonomic characters. |
| | Can DNA be extracted from a specimen used for non-destructive DNA extraction? | Yes, for most taxa this is possible. It can be difficult for small taxa like Acarina and some Diptera |
| 2. Comparison of bulk non-destructive extraction to tissue-based DNA extraction for metabarcoding samples of known composition—multiple individuals | How does bulk non-destructive (ND) extraction compares to total homogenization (H) and tissue dissection (D) based extractions when detecting taxa with metabarcoding? | Detection of taxa was similar across all three extraction methods with >84% of taxa detected. However, some more sclerotized taxa (Coleoptera) were often missed by the ND method compared to other methods. |
| | Can specimens from bulk ND extractions be used for DNA barcoding and taxonomic identification? | Yes, unlike individual ND extraction, specimens subject to bulk ND were largely unaltered and were easily used for DNA barcoding and taxonomic identification |
| 3. Non-destructive DNA extraction of rapid bioassessment samples stored for morphological identification—multiple individuals | Can taxa be detected in field samples when ND extraction is used? | Yes, most taxa were detected using ND extraction protocol except Hydrobiidae snail and some small taxa. |
| | Do storage conditions (100% ethanol with refrigeration) and sub-optimally (70% ethanol at room temperature) affect taxa detection success when using non-destructive DNA extraction? | There were no obvious difference in the number of taxa detected between samples stored under different conditions. |

barcodes from GenBank (http://www.ncbi.nlm.nih.gov/genbank/) and BOLD systems v3 (http://www.boldsystems.org/) databases from families or orders known to have aquatic life stages. The ten best matches were also checked to ensure that species matches were specific. Chimeric sequences were identified and removed with USEARCH version 9 (*Edgar et al., 2011*) using the freshwater macroinvertebrates as a reference database. Specimens from any families not detected at >97.5% with HTS were subjected to individual DNA barcoding (see above).

## RESULTS

### Component 1: individual non-destructive DNA extraction

We obtained DNA suitable for amplifying DNA barcodes from individual freshwater macroinvertebrates following a non-destructive DNA extraction (Tables 1 and 2). Sanger sequencing on amplified DNA barcodes revealed the expected species DNA. We obtained DNA barcodes for all specimens in experiment 1a (Table 2), but some Acarina and Oligochaeta in experiment 1b that amplified did not produce amplicons that could be clearly sequenced (Table 3).

For most taxa, whole specimens only required immersion, and subsequent incubation for one hour at 56 °C in the T1 buffer with proteinase K from the Nucleospin kit was sufficient to extract DNA for PCR amplification. However, we found that complete digestion of tissue in species from the Gastropoda, Oligochaeta and Hirudinea (Tables 2 and 3) was an

Carew et al. (2018), *PeerJ*, DOI 10.7717/peerj.4980

**Table 2** **Results of experiment 1a examining the effect of incubation time in the T1 buffer with proteinase K (from the Nucleospin DNA extraction kit) on non-destructive total genomic extractions of species from nine invertebrate groups** The success of non-destructive (ND) extraction (Extraction 1) for standard DNA barcoding PCR is indicated by the '+' symbol ('−' for no PCR product), while the success of the second Chelex extraction (Extraction 2) on the same material is indicated the adjacent column. GenBank accession numbers for sequenced specimens are given in parenthesis.

| Order/ subclass | Species (family) | Sclerotization | Life stage | Incubation time for Extraction 1 | | | | | |
| --- | --- | --- | --- | --- | --- | --- | --- | --- | --- |
| | | | | 30 min | | 1 hr | | 3 hr | |
| | | | | Extraction 1 (1st ND) | Extraction 2 (2nd Chelex) | Extraction 1 (1st ND) | Extraction 2 (2nd Chelex) | Extraction 1 (1st ND) | Extraction 2 (2nd Chelex) |
| Oligochaeta | *Lumbriculus variegatus* (Lumbriculidae) | Soft bodied | Adult | + (MG976202) | − | + | − | | |
| Gastropoda | *Physa acuta* (Physidae) | Soft bodied/shell | Adult | + (MG976201) | + (MG976114) | + | + | + | − |
| Amphipoda | *Austrochiltonia subtenuis* (Chiltoniidae) | Light sclerotized | Adult | + (MG976104) | + (MG976105) | + | + | | |
| Diptera | *Procladius villosimanus* (Chironomidae) | Sclerotized head | Larvae | | | + (MG976143) | + (MG976144) | + | + |
| Trichoptera | *Hellyethira simplex* (Hydroptilidae) | Sclerotized head and thorax | Larvae | + (MG976170) | + (MG976171) | + (+) | | | |
| Ephemeroptera | *Offadens* sp. (Baetidae) | Sclerotized | Nymph | + (MG976108) | + (MG976109) | + (+) | | | |
| Plecoptera | *Dinotoperla thwaites* (Gripopterygidae) | Sclerotized | Nymph | + (MG976167) | + (MG976167) | + (+) | | | |
| Hemiptera | *Micronecta* sp. (Corixidae) | Sclerotized | Adult | + (MG976178) | + (MG976178) | + (+) | | | |
| Coleoptera | *Necterosoma* sp. (Dytiscidae) | Heavily sclerotized | Adult | − | + | + (MH000193) | +[a] (MH000194) | + | +[a] |

**Notes.**

[a] Extraction 2 using crushed legs failed, but 2 hr incubation of whole animal in Chelex (with proteinase K) was successful.

**Table 3  Results of experiment 1b examining non-destructive DNA extraction trial on individuals from multiple macroinvertebrate groups.** A breakdown of the species in each family can be found in Table S1.

| Higher taxonomic rank | Sclerotization | Families | Material for extraction | | Individuals tested | Number of individuals with successful extractions | | |
|---|---|---|---|---|---|---|---|---|
| | | | Extraction 1 | Extraction 2 | | Extraction 1 (1st ND) | Extraction 2 (2nd Chelex) | Both extractions |
| Acarina | Moderate | 2[a] | Whole animal | Whole animal | 6 | 4 | 1 | 1 |
| Oligochaeta | Soft bodied | 3[a] | Whole animal | No material | 5 | 5 | 0 | 0 |
| Hirudinea | Soft bodied | 1 | Whole animal | No material | 1 | 1 | 0 | 0 |
| Bivalvia | Soft bodied/shell | 1 | Whole animal | No material | 3 | 2 | 0 | 0 |
| Gastropoda | Soft bodied/shell | 2 | Whole animal | Whole animal | 5 | 5 | 4 | 4 |
| Diptera | Little - moderate | 6 | Whole animal | Tissue sample | 12 | 11 | 5 | 5 |
| Trichoptera | Moderate | 5 | Whole animal | Whole animal/leg | 8 | 8 | 8 | 8 |
| Ephemeroptera | Moderate | 2 | Whole animal | Leg | 4 | 4 | 4 | 4 |
| Plecoptera | Moderate | 2 | Whole animal | Leg | 4 | 4 | 4 | 4 |
| Hemiptera | Moderate - heavy | 3 | Whole animal | Leg | 4 | 4 | 4 | 4 |
| Coleoptera | Moderate - heavy | 5 | Whole animal | Leg | 8 | 8 | 8 | 8 |
| Amphipoda | Moderate | 2 | Whole animal | Leg | 4 | 4 | 4 | 4 |
| Megaloptera | Moderate | 1 | Whole animal | Leg | 1 | 1 | 1 | 1 |
| Decapoda | Moderate - heavy | 2 | Whole animal | Leg | 2 | 2 | 2 | 2 |
| Odonata | Moderate | 1 | Whole animal | Leg | 1 | 1 | 1 | 1 |

**Notes.**
[a]All families in these orders were not identified. Lowest possible identifications are given.

issue. The 30 min and/or 1 hr incubation completely digested Oligochaeta and Hirudinea specimens, suggesting that these orders are unlikely to be suitable for non-destructive extraction when whole animals are immersed in extraction buffer for 30 mins or longer. The longer incubation of 3 hrs completely digested tissues in the Gastropoda, but some tissues remained after 1 hr (or 30 min), making a second extraction possible. The presence of a shell meant that taxonomic examination was also possible. We failed to obtain DNA for amplification from an adult Coleopteran after 30 min immersed in the extraction buffer but did after an incubation time of $\geq$ 1 hr (Table 2) suggesting these specimens require longer immersion in extraction buffer for DNA extraction.

Using an one hour incubation as a standard, it was possible to obtain DNA with a non-destructive total genomic extraction (Extraction 1) from a taxonomically wide range of species (Table 3). However, we found some specimens from the Acarina, Bivalvia as well as a Ceratopogonidae specimen did not amplify. Given that other specimens from these groups did amplify, it is possible that the undetected taxa were extracted successfully but were not amplified by the standard Folmer DNA barcoding primers (LCOI1490/HCOI2198).

We were also able to perform a second Chelex extraction (Extraction 2) on most specimens, including those where DNA was extracted from legs (Amphipoda, Trichoptera, Ephemeroptera, Plecoptera, Hemiptera, Coleoptera (larvae), Megaloptera and Decapoda). However, only some Diptera and Acarina were successfully amplified in the second Chelex extraction. Dissected Diptera tissues tended to stick to forceps during dissection, and this may have resulted in some material being lost. For the Acarina, all tissue with DNA may have been completely digested in the first extraction because of the small size of specimens.

Most specimens were sufficiently intact for morphological identification (Fig. S1A). Some specimens had a coating of white residue from the T1 extraction buffer, but this could be removed by adding more ethanol or by manual removal using forceps. Many of the less sclerotized specimens had a hyaline appearance, but this assisted identification when specimens were mounted, particularly for Diptera (i.e., Chironomidae). However, the Amphipoda were substantially damaged during the extraction process and were often missing antennae and legs which are needed for taxonomic identification.

## Component 2: bulk DNA extraction for metabarcoding

The three extraction methods based on homogenization, dissection and non-destructive extraction of small taxa varied in the amount of time and number of tubes required (Tables 1 and 4). The most rapid method involved completely homogenizing small taxa in samples (H) along with dissected large taxa, but this method also used the most microcentrifuge tubes for DNA extraction, increasing the cost of extraction per sample. The slowest method was sample dissection (D) of all taxa, which took 2–3 times longer than the homogenization (H) protocol, but this method meant all DNA could be extracted in a single microcentrifuge tube. Non-destructive extraction (ND) only took 1.5 times longer than the homogenization (H) protocol to complete, but used fewer microcentrifuge tubes.

All DNA extraction methods produced amplicons for each of the three HTS primer sets, with greater than 84% of the expected taxa detected in samples of known taxon composition (Table 5). The DNA extraction based on dissection of specimens (D) produced the most

**Table 4 Sample preparation and comparison of species detection using metabarcoding based on three DNA extraction methods.** Extraction methods are based on non-destructive (ND), homogenization (H) and dissection (D) in samples of known taxonomic composition containing 80, 100 or 120 individuals.

| | Extraction method | Sample A 80 individuals | Sample B 100 individuals | Sample C 120 individuals (technical replicate) |
|---|---|---|---|---|
| Time taken to prepare sample for DNA extraction (min) | ND | 10 | 15 | 10 |
| | H | 7 | 10 | 8 |
| | D | 21 | 20 | 27 |
| Number of DNA extraction tubes required | ND | 2 | 2 | 2 |
| | H | 3 | 3 | 3 |
| | D | 1 | 1 | 1 |
| Number of species detected with HTS | ND | 46 | 33 | 40 (40) |
| | H | 38 | 45 | 37(38) |
| | D | 47 | 41 | 41 (42) |
| % of expected species or genera detected with metabarcoding compared to samples of known taxonomic composition | ND | 93 | 89 | 98 (98) |
| | H | 88 | 100 | 98 (100) |
| | D | 98 | 100 | 98 (100) |

reliable detection of taxa, with only a single taxon, *Micronecta* sp., not detected in the sample containing 80 individuals (Fig. 2). Samples that were homogenized or subjected to non-destructive DNA extraction were more variably detected. Generally, homogenized samples with ≥100 individuals detected most taxa (Fig. 2). However, six species, *Cosmioperla* sp. ABX7338 (Eustheniidae), *Anisocentropus latifascia* (Calamoceratidae), *Paratanytarsus grimmii* (Chironomidae), *Micronecta* sp. (Corixidae) *Tasmanophlebia* sp. ACM3395 (Oniscigastridae) and *Diplacodes haematodes* (Libellulidae), were not detected in the sample containing 80 individuals. Similarly, up to four taxa were not detected in the samples that underwent non-destructive extraction. These often represented the more sclerotized taxa. For example, heavily sclerotized Coleopterans from the Dytiscidae, Ptilodactylidae and Elmidae were not detected in some samples (Fig. 2). We also found that some chironomids were not detected in the sample containing 100 individuals; and the Calocidae (Trichoptera), which can retract into a stone case, were not detected in samples containing 80 and 100 individuals.

Particular extraction methods performed better for some taxa in terms of the number of reads (or sequences) produced and consistency of detection (Fig. 2). For example, adult Dytiscidae (*Necterosoma* sp.) consistently produced >1,800 reads when whole adult beetles were homogenized for DNA extraction, but when legs were dissected or whole specimens were immersed for DNA extraction, <20 reads were produced. In contrast, the ND method consistently detected Corixidae (*Micronecta* sp.) with >70 reads, whereas H and D consistently produced a low number of reads, and sometimes failed to detect this taxon. Most of the remaining taxa were readily detected with all three extraction methods. Technical replication of sample C with all three extraction methods showed that metabarcoding method was robust (Fig. 2). There were only two instances of where taxon represented with low reads was detected in one replicate but not the other.

**Table 5 Amplicon size and number of reads (sequences) obtained for each of the three amplicons used for HTS.** Samples include three sets of constructed samples containing 3 replicates extracted using non-destructive (ND), complete homogenization (H) and dissection (D) based DNA extraction protocols, and six field-collected macroinvertebrate samples from three sites along Merri Creek (Melbourne, Australia) extracted using a non-destructive extraction protocol.

| Sample (number of individuals) | Extraction method | LCOI/MLepR2 amplicon | B/COIBrev amplicon | mtCOIintF/HCOI amplicon | | Overall species detection success (%) |
|---|---|---|---|---|---|---|
| | | 280 bp | 293 bp | 313 bp | Total | |
| **Samples of known composition:** | | | | | | |
| Sample A (80) | ND | 48,288 | 56,031 | 44,645 | 148,964 | 92 |
| | H | 117,667 | 116,313 | 61,850 | 295,830 | 84 |
| | D | 79,356 | 86,299 | 38,716 | 204,371 | 97 |
| Sample B (100) | ND | 67,448 | 75,433 | 19,463 | 162,344 | 89 |
| | H | 44,783 | 49,313 | 14,826 | 108,922 | 100 |
| | D | 111,207 | 124,103 | 68,281 | 303,591 | 100 |
| Sample C (120) | ND | 76,164 | 54,706 | 32,963 | 163,833 | 95 |
| | H | 39,418 | 65,326 | 24,561 | 129,305 | 95 |
| | D | 79,524 | 109,656 | 26,218 | 215,398 | 95 |
| Sample C (120) technical replicate | ND | 70,875 | 65,871 | 20,231 | 156,977 | 95 |
| | H | 86,622 | 134,934 | 24,233 | 245,789 | 100 |
| | D | 50,590 | 93,821 | 26,826 | 171,237 | 100 |
| **Field collected samples:** | | | | | | |
| MRD 2016 | ND | 75,891 | 44,593 | 69,640 | 190,124 | 88 |
| MRD 2014 | ND | 69,445 | 43,188 | 57,083 | 169,716 | 73 |
| MOH 2016 | ND | 47,943 | 41,761 | 57,124 | 146,828 | 83 |
| MOH 2014 | ND | 67,644 | 49,726 | 39,482 | 156,852 | 80 |
| MCL 2016 | ND | 81,108 | 15,177 | 60,074 | 156,359 | 73 |
| MCL 2014 | ND | 727 | 687 | 164 | 1,578 | 55 (low reads) |

Overall, when multiple specimens were combined for non-destructive DNA extraction in the T1 buffer with proteinase K and incubated for 1 hr, there was less digestion of tissues in specimens when compared to digestion in the individual DNA extraction trials (Fig. S2). In particular, amphipod specimens were largely unaltered by bulk non-destructive extraction. After non-destructive DNA extraction, several taxa were successfully amplified and DNA sequenced following a second Chelex extraction (Table S3).

## Component 3: detection of macroinvertebrate taxa in field samples

A high number of sequences were obtained for each of the field-collected samples from Merri Creek, with the exception of the 2014 Coburg Lake (MCL 2014) sample (Fig. 3). Overall, most taxa were detected in samples, including the 2014 Coburg Lake sample (Table 5). Species from the orders Gastropoda, Diptera, Coleoptera, Ephemeroptera, Hemiptera and Trichoptera were detected through the ND protocol (Table 1; Fig. 3). However, we found that gastropods from the family Hydrobiidae (which have an operculum) were consistently missed with this protocol. We also encountered some individual taxa that were not detected in samples. These tended to be small specimens, such as the Hydroptilidae in the 2016 Coburg Lake sample (MCL) and the Tanypodinae

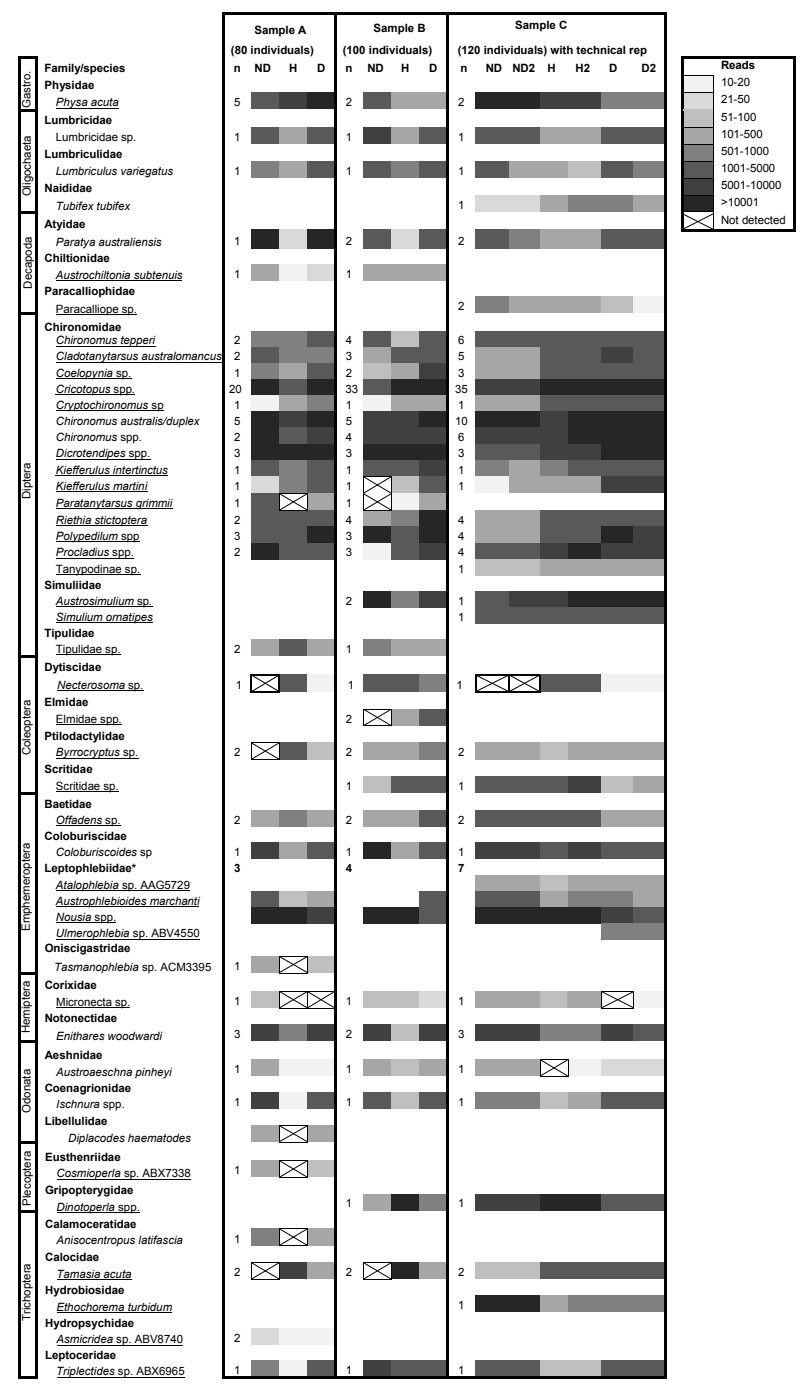

**Figure 2** **High throughput DNA sequencing of nine constructed samples of known taxonomic composition extracted using non-destructive (ND), complete homogenization (H) and dissection (D) based DNA extraction protocols.** Samples contain 80, 100 or 120 individuals. Sample C was run with a technical replicate. The greyscale indicates the number of reads (sequences) returned that match each taxon and 'n' the number of specimens used from each taxonomic group to compose samples. Species underlined were 'small taxa' extracted using the non-destructive method. Note: the Leptophlebiidae could not be easily distinguished as many specimens were missing legs and were early instars, so detection within samples with the same number of individuals varied.

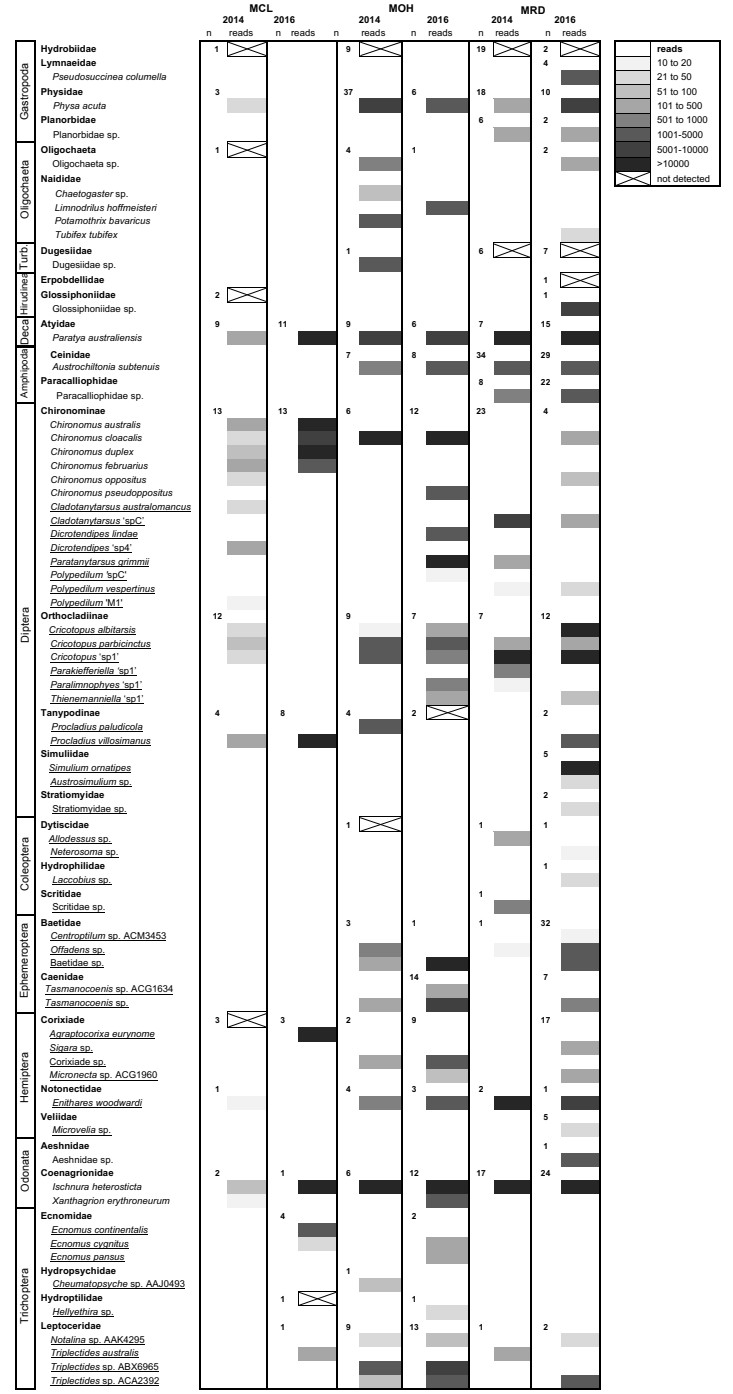

**Figure 3** **High-throughput DNA sequencing of six field-collected samples from Merri Creek 2014–2016.** Sites in Melbourne, Australia include Merri Creek at Coburg Lake, Coburg (MCL); Merri Creek at O'Herns Road, Broadmeadows (MOH) and Merri Creek at Rushwood Drive, Craigieburn (MRD). The greyscale indicates the number of reads (sequences) returned that match each taxon and 'n' the number of individuals morphologically identified from each family. Species underlined were 'small taxa' extracted using the non-destructive method. Some species with unclear taxonomy are listed by their BOLD BIN (three letters and four numbers) which can be used to find the sequence data and taxonomic information on the taxon on the BOLD version 3 website at http://v3.boldsystems.org/.

 

in the 2016 O'Herns Rd (MOH 2016) sample, or more sclerotized specimens such as the Corixidae in the 2014 MCL sample and the Dytiscidae in the 2014 MOH sample. Some non-detections likely resulted from amplification failure rather than extraction failure; e.g., some non-insect taxa, involving species of Oligochaeta, Dugesiidae and Hirudinea, were not detected. There were no obvious differences in the likelihood of detection of species in the 2014 samples compared to 2016 samples (Fig. 3), even though the latter were better preserved for DNA isolation after being stored in 100% ethanol at 4 °C after collection rather than in 70% ethanol at room temperature.

Specimens subjected to the non-destructive extraction protocol remained suitable for subsequent DNA barcoding and taxonomic examination, as established through DNA amplification and sequencing of the DNA barcode region and the presence of taxonomic characters.

## DISCUSSION

Our study showed that non-destructive DNA extraction could be used for preparing invertebrate samples for metabarcoding. It can allow many different taxa from samples used for metabarcoding to also be used for individual DNA barcoding and taxonomic examination, particularly at the larval/nymph stage. This can help reference DNA barcoding for species level identification, an important step given that incomplete DNA barcoding reference libraries remain a key limitation to identifying species when conducting metabarcoding of environmental samples (*Aylagas, Borja & Rodriguez-Ezpeleta, 2014*; *Cristescu, 2014*; *Carew et al., 2017*; *Elbrecht et al., 2017*). In addition, using a non-destructive extraction method can facilitate 'more targeted' production of DNA barcoding reference libraries, particularly where samples have first been identified to coarser taxonomic levels (i.e., family/genus), such as those from rapid bioassessment surveys (i.e., *Carew et al., 2016*; *Elbrecht et al., 2017*) as it enables specimens to be examined after they are metabarcoded. In this case, metabarcoding can be conducted first and specimens without DNA barcodes can be recognized later based on morphological examination. These taxa can be removed from samples and be targeted for individual DNA barcoding and further taxonomic investigation to build DNA barcode library coverage. This process is particularly useful for areas where routine surveys are conducted or for sites which are repeatedly surveyed, leading to locally comprehensive DNA reference libraries for determining species diversity. With further testing, non-destructive extraction may become particularly useful when invertebrate samples cannot be destroyed for metabarcoding, such as those in archived or museum collections (i.e., *Carew et al., 2016*).

While more reliable detection of biodiversity (see *Elbrecht & Leese, 2017*) may be possible with DNA extraction protocols using complete homogenization and replication of the extraction step, we did not to compare the effect of replication at the extraction step in this study. Therefore, it was unclear whether replication would improve the detection of taxa using non-destructive extraction. However, we did complete technical replication of one sample for each extraction method when metabarcoding. This revealed that our metabarcoding was highly reproducible, with same taxa detected in each replicate when
using non-destructive extraction. While the other two extraction methods failed to detect one taxon represented by a low number of reads between replicates. This showed that for this sample replication of the metabarcoding step was not important for increasing detection of biodiversity. However, replication of more samples would be useful to determine if this is always the case.

Overall, species detection success for the extraction methods trialled in this study were similar to those employed by other metabarcoding studies using invertebrate samples (*Gibson et al., 2014*; *Elbrecht & Leese, 2015*; *Carew et al., 2016*; *Elbrecht, Peinert & Leese, 2017*; *Elbrecht et al., 2017*). We found the number of species detected with metabarcoding in samples prepared using non-destructive DNA extraction were similar to those where small taxa were dissected or whole individuals homogenized. Moreover, large taxa which were processed via homogenization in all extraction methods were mostly detected at a similar rate to those prepared using non-destructive DNA extraction.

However, where we did not detect taxa, there was bias towards particular groups for the three extraction methods tested here. For example, heavily sclerotized taxa such as Coleoptera were readily detected when whole individuals were homogenized, but often failed to be detected when subjected to non-destructive DNA extraction. It is likely more DNA is released by homogenization of the whole animal, resulting in more DNA being available for PCR. Typically, species biomass is linked to detection success when metabarcoding, and large or more common species can affect the detection of smaller rarer species that contribute less DNA (*Elbrecht & Leese, 2015*; *Dowle et al., 2016*; *Elbrecht, Peinert & Leese, 2017*). PCR biases, due to PCR primer selection and the number of primer sets used, can also lead to metabarcoding failing to detect some taxa from particular orders or families (*Clarke et al., 2014*; *Brandon-Mong et al., 2015*; *Aylagas et al., 2016*; *Elbrecht & Leese, 2017*). Our lower detection rates in field samples probably occurred because of a lack of degeneracy in metabarcoding DNA primers, especially in the case of non-insect taxa, such as the Oligochaeta, Hirudinea and Dugesiidae. Additional primer sets, such as those suggested in *Elbrecht et al. (2017)* which are specifically designed for freshwater invertebrates, could improve the detection of these taxa.

While non-destructive DNA extraction did create some amplification bias when metabarcoding, the method could be modified to improve detection of certain taxa. For example, more sclerotized taxa (adult and some larval Coleoptera), or taxa retracted into a stone case (some Trichoptera e.g., Calocidae) or having shell with an operculum (Gastropoda e.g., Hydrobiidae), were less likely to be detected. To counter this, a longer incubation time in the T1 extraction buffer with proteinase K may be needed during bulk non-destructive extraction to release sufficient DNA for metabarcoding. Therefore, it may be necessary to separate these taxa from less sclerotized specimens and then use an incubation time >1 hr for non-destructive DNA extraction. Based on initial trials, up to 3 hr may be suitable for these animals, as DNA barcodes were successfully amplified when an individual non-destructive DNA extraction was performed. While this would increase the time taken to conduct DNA extractions due to the requirement for coarse sorting, it would likely be more rapid than dissecting specimens to preserve taxonomic features. Further testing and replication considering a broader range of taxa would be useful to refine

the non-destructive DNA extraction protocol for routine use, especially where validation of metabarcoding and individual DNA barcoding for reference libraries are still required.

There was a substantial difference in appearance of specimens extracted non-destructively as individuals compared to those extracted in bulk. For example, most Diptera and Amphipoda were morphologically unaltered after bulk extraction but had a strong hyaline appearance after individual-based extraction. The hyaline appearance likely reflected a high degree of tissue digestion after 1 hr of incubation in the extraction buffer. In contrast, in most other sclerotized taxa, bulk versus individual extraction had little impact on detection and the former approach often left specimens in better condition for individual DNA barcoding and taxonomic examination. While dissection of all taxa in samples also allowed for taxa to be re-examined, this took substantial time and often led to some taxa being damaged. In particular, small Diptera and Gastropoda were not easily dissected, often resulting in damaged taxonomic characters needed for identification. The morphological integrity of the smaller taxa was better preserved using non-destructive DNA extraction.

Non-destructive DNA extraction of small taxa in combination with dissection of large taxa, and those unsuited to non-destructive DNA extraction (e.g., Oligochaeta, Hirudinea and Dugesiidae), enabled DNA barcoding of taxa with varying levels of sclerotization from over 15 different macroinvertebrate orders. This included invertebrate samples that were stored for at least three years in 70% ethanol at room temperature which is a common practice when morphological examination is needed (*Rosenberg & Resh, 1993*; *Haase et al., 2004*), and samples stored in 100% ethanol at 4 °C which provides better preservation of DNA (*Baird et al., 2011*; *Stein et al., 2014*). This means the non-destructive approach could be trialed for extracting DNA from archived or museum collections, even if invertebrates have been stored in 70% ethanol for up to three years. However, taxon detection becomes problematic in samples stored for greater than five year in sub-optimal conditions for DNA preservation (i.e., *Carew et al., 2016*). Therefore, further testing of non-destructive DNA extraction would be useful to determine how it performs on older material.

## CONCLUSIONS

In summary, we show that non-destructive DNA extraction protocols can be used for preparing a variety of freshwater invertebrate species for bulk DNA extraction and subsequent metabarcoding. When non-destructive DNA extraction of small taxa is combined with dissection of large taxa, detection of species diversity is comparable to other DNA extraction methods. With further refinement the approach offers means to increase the speed of bulk DNA extraction of invertebrate samples for metabarcoding, while enabling the same samples to be used for individual DNA barcoding and taxonomic identification. The approach also appears suitable for samples not specifically stored for DNA-based approaches.

## ACKNOWLEDGEMENTS

The authors would like to thank Claudette Kellar, David Sharley, Steve Marshall, Eddie Tsyrlin and Cameron Amos for providing macroinvertebrate samples for this study.

### Funding

All funding for this study was from the Australian Research Council through their grant (grant number LP150100876) and Fellowship (grant number FL100100066), with additional support from Melbourne Water Corporation. There was no additional external funding received for this study. The funders had no role in study design, data collection and analysis, decision to publish, or preparation of the manuscript.

### Grant Disclosures

The following grant information was disclosed by the authors:
Australian Research Council: LP150100876, FL100100066.
Melbourne Water Corporation.

### Competing Interests

The authors declare there are no competing interests.

### Author Contributions

- Melissa E. Carew conceived and designed the experiments, performed the experiments, analyzed the data, contributed reagents/materials/analysis tools, prepared figures and/or tables, authored or reviewed drafts of the paper, approved the final draft.
- Rhys A. Coleman and Ary A. Hoffmann conceived and designed the experiments, contributed reagents/materials/analysis tools, authored or reviewed drafts of the paper, approved the final draft.

### DNA Deposition

The following information was supplied regarding the deposition of DNA sequences:

DNA sequences from metabarcoding of constructed and field samples are under NCBI BioProject: PRJNA413851 and the MiSeq data is entered in NCBI SRA with the following accession numbers: SAMN07768647–SAMN07768661.

GenBank Accessions for Cytochrome oxidase I are MG976085–MG976217 and MH000193–MH000194.

Fasta files: CAREW, MELISSA (2018): freshwater_macroinvertebrate_DNAbarcodes.fasta. University of Melbourne. Dataset. https://doi.org/10.4225/49/5b06794a1addf.

### Supplemental Information

Supplemental information for this article can be found online at http://dx.doi.org/10.7717/peerj.4980#supplemental-information.

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
