# Peer review of "Can non-destructive DNA extraction of bulk invertebrate samples be used for metabarcoding?"

_PeerJ, doi:10.7717/peerj.4980_

## Round 0.1 · original submission · Major Revisions

Dear Melissa,

I have now received three very positive reviews of your study (all minor revisions), well done! All reviewers have acknowledge the quality and importance of this work, but have still highlighted a number of minor issues in relation to the introduction, flow of experimental design and the requirement for additional discussion points. For this reason, I am requesting a major revision, but is is highly likely that this study will be considered positively for publication in PeerJ once all issues have been incorporated into a revised manuscript and a point-by-point rebuttal provided.

I will be looking forward to receiving your revised manuscript.

With kind regards,
Xavier

·

Basic reporting

No comment

Experimental design

No comment

Validity of the findings

I think one important point that could also be discussed is whether you observed any issues with cross-contamination when trying to amplify/sequence individual specimens post bulk non-destructive extraction. To my mind, this would be one of the major confounding factors for re-using specimens after bulk treatment – i.e., residual mixed DNA from the bulk extraction retained on each individual body that might result in co-amplification of barcodes in what should be an individual reaction. If this was not observed to be a problem, that is worth mentioning in the results and discussing at greater length. Especially given the comment early in the paper on the ability to use the ethanol from sample vials for metabarcoding - akin to an eDNA approach.

Additional comments

The paper is well written, clear, concise, all figures and tables are easily interpretable, and provides sufficient introduction to concepts and citation of relevant literature. I have no problem recommending this for publication pending some very minor changes.

Minor comments
Introduction – maybe worth including a citation for Krosch & Cranston (2012, Chironomus)? Authors can decide relevance – this paper reported a modified protocol for non-destructive DNA extraction from chironomid pupal skins that expands the barcode-able collection of biomonitoring programs.
Line 108 – ‘for intended for’, remove first ‘for’
Line 226 – change ‘primers’ to ‘primer’
Line 392 – change ‘such those’ to ‘such as those’
Line 394 – remove ‘as’
Line 402 – ‘trialled’
Line 460 – change ‘i.e.’ to ‘e.g.’
Table S1 – amend column widths to avoid split words (maybe landscape layout would be better). Some genus names are italicised, some not – please make consistent

·

Basic reporting

The paper by Carew et al. aims to test if freshwater macroinvertebrates used for non-destructive DNA-extraction are damaged beyond subsequent species recognition, and if DNA can be extracted in a second round. The findings are interesting to a wide audience and of importance for future development of DNA metabarcoding for monitoring and assessment of freshwater ecosystems. Although I have heard people talk about trials at conferences, I have not seen any published studies presenting this proof-of-concept before.

The paper is written in a clear and concise language and I think the authors do a very good job in presenting the results from a complex experimental setup. The use of references is very good and the background information sufficient for the reader to get a relevant impression of the field.

The use of figures and tables is extensive, but adds value to the paper by presenting results in a clear and easy-to-understand manner.

Experimental design

The experimental design is quite complex, but makes sense in the light of the primary research questions asked. The most problematic issue is the lack of true replicates to evaluate the variance often observed in multiple extractions of the same bulk sample. I know this can be resource demanding to achieve, but the good thing is that the authors have one technical replicate in experiment 2 (with 120 individuals).

I am wondering if residue ethanol in the bulk sample before adding lysis buffer might have influenced the DNA extraction success of "hard" animals. In the methods the authors describe that ethanol was pipetted out, but not if the sample was dried completely before buffer was added. If residue ethanol was present inside beetles and other hard shelled animals, it would prevent lysis and extraction from these specimens.

Validity of the findings

The discussion is mostly adequate and good, but should include considerable more detail on the issues mentioned under experimental design. The lack of replicates reduces the explanatory power of the results and this should be mentioned. Also, there are some differences observed between the two replicates that the authors do present. Please discuss this with reference to papers that argue for multiple DNA extractions of the same bulk sample when doing metabarcoding.

Please also discuss the eventual influence of the ethanol removal step in the extraction of DNA from bulk samples.

·

Basic reporting

Some editing is required through the text. It is hard to follow, some sentences are weak and the way in which they are written makes the paper very dense. I understand that as a methodological paper, many details need to be provided. However, I recommend some editing to make some parts more concise. For example, line 235: …., and were amplified in triplicate using…. [skip ‘amplifications were performed as three replicates’]. The PCR replicates were pooled and cleaned…

Also, the flow of the procedure is very confusing. I found quite hard following the direction of the experiment (Component 1). I suggest the authors re-writing this section following a unique direction from Extraction 1 to Extraction 2, without going back again to Experiment 1 and then to Experiment 2 many times. Also, to be consistent between Fig 1 and Table 1, consider renaming Experiment 1a and Extraction 1; and Experiment 1b and Extraction 2 using the same term. It is quite difficult to follow. In addition, I recommend moving Tables 2 and 3 to the results section.

In contrast, in Component 3 more detail in the text is needed. Many steps depicted in the figure are not explained and makes it difficult to follow.

Figure 1. Add accordingly a, b, c because it is mentioned in the text but not in the figure.

Table 1. As mentioned above, try to be consistent with the nomenclature between Table 1 and Figure 1 to ease the flow. Also, the information provided by Aims and Question is redundant. To make the table more informative, I recommend the authors, skip one of them and add a third column with the answer to each question/aim.

Experimental design

I thank the authors such a detailed experimental design to address many questions crucial for the better understanding of metabarcoding methods for environmental monitoring. I recognize the big amount of work that this experiment requires and the detailed manipulation of animals needed to mimic the samples (especially in Component 2). Sufficient detail is provided (except for Component 3 - see comment above)

The questions of the paper are clear enough but the introduction lacks a sentence with the hypothesis/aims/questions of this research. Which then can be followed by the procedures in Line 86-96

Validity of the findings

The findings regarding the amplification success greater than 88% (Line 329) is not reflected in the material indicated for that (table 5). Table 5 only provides the number of reads obtained for each sample of known composition, but not for the amplification success. Please provide.

The findings regarding Component 3 must be supported with figures/tables. The figure supporting the statement that "Overall, most taxa were detected in samples (Line 361-362)" must be provided/mentioned.

The discussion is well stated and linked to the original question. I highlight the fact that the possible reasons for some failures in the method are discussed. Yet, I highly recommend extending the discussion focussing on the pros and cons of this method in the case it is aimed to be implemented in routine monitoring. If so, a high economic investment (cost of consumables and time) would be required. An attempt is done in lines 321-323, but it should be better explained.

Additional comments

I congratulate the authors for the nice experimental design of this research. I provide some minor comments that should be addressed.

It is expected that the digestion in pk in Extraction 1 will degrade some tissues making it difficult for the subsequent taxonomic identification of specimens. That might be problematic if, in practice, this procedure is followed. Wouldn’t be possible swapping the order of the steps and performing the morphological identification first and then going for the DNA extraction? Clarify this, please.

Move lines 113-115 to the beginning of methods section

Line 132: specify the target gene (COI). Also, it is well known that the degenerated versions of this primer set perform better than the Folmer. Why did not the authors use those?

Explain what sort of technical replicates (i.e. metabarcoding samples twice).

Line 221: this sentence leads to misunderstanding. It appears that the DNA barcode region has only three amplicons. Please re-write to give sense to the sentence.

Line 228-230: move to line 240. It interrupts the flow as you are still describing the first round of PCR.

Line 266-268: specify the search in the databases. All barcodes belonging to freshwater invertebrates?

Line 279: Fig 1a shouldn’t take place here; it is only methodology.

Table S1: provide it in a different format so that the test in the column fits properly.

Table S2: From 80, 100 and 120 individuals in Sample A, B and C, only 2, 8 and 6, respectively, could be DNA barcoded? Explain the reason.

---

## Round 0.2 · Minor Revisions

Dear Melissa,

I have carefully reviewed your rebuttal and revised manuscript, and am happy with the way you have handled the reviewers' comments. However, I have found a number of small mistakes throughout the revised manuscript (see "Editor_Annotated" ms attached), and will need you to incorporate them before I can accept the final manuscript.

With kind regards,
Xavier

---

## Round 0.3 · accepted · Accept

Dear Melissa and co-authors,

I am pleased with your last modifications and am delighted to accept this study for publication in PeerJ. This work required considerable efforts on your part and I commend you for your hard work and perseverance.

Concerning the reference database you assembled/used in the study, I strongly feel it should be made available to any researcher wanting to replicate your work or build on it. You mentioned that the size of the file is 800MB - this is not big and could easily be uploaded in a sharing platform such as e.g. https://figshare.com with a link provided in the publication.

Well done and thanks again for this awesome contribution to the field.

Kind regards,
Xavier

#